# Radiotherapy-Induced High Neutrophil-to-Lymphocyte Ratio is a Negative Prognostic Factor in Patients with Breast Cancer

**DOI:** 10.3390/cancers12071896

**Published:** 2020-07-14

**Authors:** Chang Ik Yoon, Dooreh Kim, Sung Gwe Ahn, Soong June Bae, Chihwan Cha, Soeun Park, Seho Park, Seung Il Kim, Hye Sun Lee, Ju Young Park, Joon Jeong

**Affiliations:** 1Department of Surgery, St Mary’s Hospital, The Catholic University of Korea, College of Medicine, Seoul 06591, Korea; fayn@hanmail.net; 2Department of Surgery, Gangnam Severance Hospital, Yonsei University College of Medicine, Seoul 06273, Korea; rlaenfpd@yuhs.ac (D.K.); asg2004@yuhs.ac (S.G.A.); mission815815@yuhs.ac (S.J.B.); chachihwan@gmail.com (C.C.); pikasonii@yuhs.ac (S.P.); 3Department of Surgery, Severance Hospital, Yonsei University College of Medicine, Seoul 03722, Korea; psh1025@yuhs.ac (S.P.); skim@yuhs.ac (S.I.K.); 4Biostatistics Collaboration Unit, Yonsei University College of Medicine, Seoul 03722, Korea; hslee1@yuhs.ac (H.S.L.); jystat@yuhs.ac (J.Y.P.)

**Keywords:** breast cancer, neutrophil-to-lymphocyte ratio, radiation therapy

## Abstract

Radiotherapy (RT) is the standard of care following breast-conserving operation in breast cancer patients. The neutrophil-to-lymphocyte ratio (NLR) reflects the systemic change caused as a result of the radiotherapy. We aimed to evaluate the association between RT and the change in NLR following the receipt of RT, and to investigate the prognostic impact. We retrospectively reviewed NLR values of breast cancer patients taken before the administration of the first and the last session of RT. The cut-off point for the NLR was determined using the Youden index and receiver operating characteristic (ROC) curve within the training set. Recurrence-free survival (RFS), distant metastasis free survival, and overall survival were the main outcomes. Patients with an NLR higher than 3.49 after RT were classified to an RT-induced high NLR group and showed a significantly higher recurrence rate compared to those with low NLR (*p* < 0.001). In a multivariate Cox proportional hazards model, RT-induced high NLR remained a significant prognostic factor (HR 2.194, 95% CI 1.230–3.912, *p* = 0.008 for tumor recurrence. We demonstrated that an increase in NLR over the course of RT has a negative impact on survival, putting these patients with RT-susceptible host immunity at a higher risk of tumor recurrence.

## 1. Introduction

The tumor immune environment plays an important role in tumor progression by promoting angiogenesis, metastasis, proliferation, and by interfering with the response to systemic treatment [1,2,3]. Neutrophils and lymphocytes play a crucial role in tumor inflammation [4]. The neutrophil-to-lymphocyte ratio (NLR), which represents the ratio of the peripheral circulating neutrophil and lymphocyte count, reflects the imbalance between immune surveillance and tumor progression [5,6,7,8,9]. Previous studies have indicated that a high NLR reflects an abnormal host-immune surveillance status, which might significantly contribute to tumor proliferation, invasion, and metastasis [10,11]. Furthermore, an elevated peripheral NLR has been identified as an independent and easy to measure prognostic biomarker for several types of cancers, including colorectal, breast, and lung [12,13,14,15,16,17,18,19]. A growing body of evidence suggests that NLR could be a predictive biomarker of systemic therapy, such as neoadjuvant chemotherapy [20,21,22,23].

Most published studies have reported that NLR can be used as a prognostic/predictive biomarker in the pre-treatment or systemic treatment setting. However, information on NLR as a prognostic factor in cancer patients receiving radiation therapy is limited. Lymphopenia has long been known to be a sequelae of radiotherapy (RT), and irradiation is also known to reduce lymphocyte function [24,25,26,27,28]. Based on these findings, we hypothesized that RT could induce the elevation of NLR, and patients with increased NLR are likely to have a poor prognosis.

In breast cancer, RT is one of the major therapeutic modalities. Particularly, irradiation of the remaining breast is standard treatment after breast conserving surgery (BCS). The aim of this study was to investigate the relationship between RT-induced high NLR and survival outcomes in patients with breast cancer, retrospectively.

## 2. Results

### 2.1. The Determination of the NLR Cut-Off Point

A total of 1044 patients fulfilled the inclusion criteria. The patients were divided into training and validation sets, in a 7:3 ratio (Appendix A, 731 and 313 patients, respectively). When the cut-off point for NLR in the training set at T2 was 3.49, the sum of sensitivity and specificity was the maximum. The AUC, sensitivity, and specificity were 0.5726, 59.6%, and 60.83%, respectively (Appendix A). Despite the low value of the AUC, the analysis was carried on sequentially with the training set and the validation set, and the cut-off point found from the Youden index demonstrated clinically meaningful results. In the training set, patients with a high NLR (≥3.49) had a poor RFS compared to those with a low NLR (Figure 1A; HR 2.14; 95% CI 1.231–3.72, *p* = 0.0058). In the validation set, NLR (≥3.49) was also significantly associated with reduced RFS (Figure 1B; HR 3.775; 95% CI 1.771–8.044, *p* = 0.0006).

### 2.2. Characteristics of Patients with RT-Induced High NLR

A total of 248 patients with NLR ≥ 3.49 at T1 were excluded from the analysis (Appendix A). The median age of the patients included in the analysis (*n* = 796) was 49 (range: 24–82) years. Of these, 284 (35.7%) patients had RT-induced high NLR and 512 (64.3%) had low NLR after RT (Appendix A). Clinical characteristics between the patients in these two groups are compared in Table 1. RT-induced high NLR was associated with LN metastasis, receipt of chemotherapy, and regional nodal irradiation (RNI). Patients with higher tumor burden were more likely to be in the RT-induced high NLR group. However, it was not related to tumor size, age, nuclear/histologic grade, ER/PR status, HER2 status, and subtype. Also, the distribution of chemotherapy regimen was similar between the groups within the patients who had received chemotherapy (data shown at Appendix A).

### 2.3. Prognostic Impact of RT-Induced High NLR

At a median follow-up time of 95 months (range: 2–151), 62 recurrence events had occurred. In addition, there were a total of 34 deaths and 54 distant metastasis events. The patients with RT-induced high NLR showed a significantly decreased RFS compared to those with low NLR (Figure 2A; HR 2.915; 95% CI 1.731–4.909, *p* < 0.0001). They also had a significantly poorer OS and DMFS compared to those with low NLR (Figure 2B; HR 2.64; 95% CI 1.306–5.339, *p* = 0.0037 and Figure 2C; HR 2.889; 95% CI 1.653–5.049, *p* < 0.0001, respectively).

In the univariate Cox proportional hazard model, NG, ER, PR, LN metastasis, receipt of chemotherapy, RNI, and RT-induced high NLR (Table 2; HR: 2.918, 95% CI: 1.750–4.866, *p* < 0.001) were found to be significant prognostic factors for RFS (Table 2). Using these factors in the multivariate Cox proportional hazard model revealed that RT-induced high NLR was a significant independent prognostic factor for RFS (Table 2; HR: 2.194, 95% CI: 1.230–3.912, *p* = 0.008).

Uni- and multivariate analyses of DMFS were performed. RT-induced high NLR was a significant prognostic factor for a short DMFS (Table 3; HR: 2.313, 95% CI: 1.299–4.120, *p* = 0.004), and a poor OS (Appendix A; HR: 2.643, 95% CI: 1.335–5.236, *p* = 0.005) in the univariate analysis. Similar trends were seen in the multivariate analysis (Appendix A; HR: 2.394, 95% CI: 1.106–5.181, *p* = 0.027). RT-induced high NLR was an independent prognostic factor along with lymph node metastasis for survival outcome.

## 3. Discussion

We believe that this is the first study to correlate NLR with clinical outcomes in breast cancer patients treated with radiation therapy after BCS. The NLR value increased following RT in all cases, both in the high and low NLR groups, with the mean NLR increasing from 1.81 to 3.31 (+82.9%). The patients with RT-induced high NLR (mean NLR increasing from 1.98 to 4.90, +147%) had greater changes in the NLR value compared to patients with RT-induced low NLR (average NLR increasing from 1.71 to 2.43, 42%). This finding indicates that the difference in NLR is incrementally associated with decreased RFS in breast cancer patients.

Many previous studies have consistently demonstrated that high NLR is associated with poor survival in patients diagnosed with various solid cancers [7,8,9,12,13,14,15,19]. While the precise mechanism and the direct role of increased NLR in early tumor recurrence are not fully understood, it has been suggested that NLR may be an indirect indicator of the host immune status. Neutrophils have both pro- and antitumor effects depending on the context. Neutrophils promote tumor progression by releasing factors that remodel the extracellular matrix in the tumor microenvironment and act on tumor cells to enhance proliferation and invasion [29]. Tumor-associated neutrophils (TANs) are also involved in the antitumor immune response and interact with other immune cells such as CD8+ T cells [30]. Preclinical studies have demonstrated that RT-induced TAN exhibits an increased production of reactive oxygen species and apoptosis of tumor cells [31]. Further, a recent study suggested that the peripheral neutrophil count correlates with the clinical outcome. A higher absolute neutrophil count was associated with lower rates of local control of the disease, and attributed to tumor resistance in mediating radiation response [32].

Radiation-induced lymphopenia elucidates the association between high NLR after the receipt of RT and poor survival outcome. Lymphocytes are highly radiosensitive, and in vitro studies have shown that the lethal dose required to reduce lymphocyte counts by 50% is approximately 2 Gy [33]. Although radiation for the breast is delivered only to the affected area of the breast and/or axilla, modeling studies have shown that the circulating peripheral blood cells are continuously exposed as the blood flows [26]. Consistent with previous studies [24,27,28,34], Yovino et al. [26] have found that the number of fractions and the planned target volume size are more likely to affect the dose of radiation delivered to the circulating cells than the dose rates or the delivery techniques in patients with malignant gliomas. In breast cancer, evolving techniques have introduced accelerated partial breast irradiation (APBI). APBI can be a promising alternative to whole breast irradiation to mitigate the inadvertent toxicity of radiation by reducing the planned target volume size in selected patients. Additionally, a recently published systematic review by Venkatesulu et al. [35] surmises that the lack of the compensatory surge of cytokines IL-7 and IL-15 as a result of homeostatic response after RT prolongs the lymphopenia to chronic phase, subsequently leading to impaired clonal expansion and maturation of lymphocytes. A Phase I study is underway to determine if the administration of exogenous IL-7 to restore the lymphocyte population can be helpful after RT (NCT02659800). Taken together with our results, we speculate that the high NLR after RT reflects immune status precisely, may play an essential role in the recurrence of breast cancer. 

In this study, we demonstrated that the high NLR group had a significantly poor survival outcome; RFS, DMFS, and OS. RNI also showed a significant association regarding RFS, DMFS, and OS. However, RNI did not remain significant when adjusted with other variables for multivariable analysis in all three outcome measures. This finding implies that the extent of target volume alone cannot predict the clinical outcome mediated by systemic changes after RT. Although we did not present the dose-volume histogram parameters, NLR can surpass the possible effect of myelosuppression caused by irradiated thoracic bone structures (scapulae, clavicles, and ribs) by identifying the patients at risk of poor clinical outcome. It is most likely due to the fact that NLR is a surrogate marker that reflects both lymphopenia and elevated neutrophil count. 

In most studies, the cut-off value is determined using NLR before the treatment [11,15,17,18,19,23,35]; we decided to set the reference sampling at T2 to plot ROC curve because we wanted to examine the subsequent changes of NLR after RT. NLR rapidly manifests the shift of hematologic biomarkers; hence, we propose that sampling around the last session of RT is sufficient to demonstrate the dynamic difference. In addition, we rigorously selected NLR from a short time period, which exhibits the changes in immune parameters in response to RT only.

This study has some limitations, the first being the retrospective design of the study. Our findings need to be confirmed in a prospective cohort. Second, the neutrophil and lymphocyte counts may vary depending on the individual’s physiological condition. We tried to minimize the effects of other factors as much as possible, but even tamoxifen has side effects associated with neutropenia. Since tamoxifen can cause neutropenia and subsequently influence the NLR value [36,37], we estimated survival in the subgroup of patients who had been exposed to tamoxifen as an adjuvant endocrine therapy. The RT-induced high NLR group had a significantly worse RFS regardless of selective estrogen receptor modulator (SERM) use (Appendix A; patients taking SERM; HR 5.55; 95% CI 1.58–19.49, *p* = 0.0144 and Appendix A; patients not taking SERM; HR 3.702; 95% CI 1.511–9.071, *p* = 0.0024). Moreover, the duration of tamoxifen exposure was short in most cases. We sought to demonstrate that NLR is less likely to be affected by tamoxifen use and that survival did not differ by tamoxifen use. Third, there is no consensus among different studies on the cut-off value for NLR, time of blood sampling, or the method employed to calculate NLR. The ROC curves and median values have been used in most studies to determine this value. The ROC curve has the advantage of providing a better discrimination value, as it can calculate the maximum sum of sensitivity and specificity. The optimal cut-off value, however, needs further investigation. A meta-analysis covering 38 datasets of 7065 patients on the impact of NLR on the outcome for various solid tumors shows a wide range of NLR cut-off values, and about 25% of studies use similar cut-off values to those in our study [38]. Lastly, the timing of blood sampling was not identical among patients, leading to variations of the interval between the actual completion of RT and the day of blood sampling as T2 (7 days, interquartile range 2–11). Therefore, the patients might have been given different doses of radiation at the time of blood sampling. Despite these limitations, we have obtained relevant evidence to show the association between RT and NLR in a large number of patients, providing insight into changes in the radiation-related immune status and survival outcomes in patients with breast cancer.

## 4. Materials and Methods

### 4.1. Patients

This retrospective study analyzed data of patients who received histological diagnosis of primary breast carcinoma from January 2006 to December 2011. We enrolled patients who were 18 years or older at the time of the surgery with histologically confirmed invasive breast cancer (stage I to III) and were undergoing BCS with adjuvant RT at the Gangnam Severance Hospital and Severance Hospital, South Korea. Of the enrolled patients, those who had ductal carcinoma in situ only, stage IV breast cancer, an active infection, an acute/chronic inflammatory disease, an autoimmune disease, a hematologic disorder except anemia, had been taking immunosuppressive medicines, and had a history of previous irradiation were excluded from the study. In addition, patients who received adjuvant RT at other hospitals were also excluded from the study to unify the radiation protocol used.

The clinical data, including age at the time of surgery, nuclear grade (NG), histologic grade (HG), tumor size, lymph node (LN) status, estrogen receptor (ER) status, progesterone receptor (PR) status, human epidermal growth factor receptor-2 (HER2) status, white blood cell (WBC) count, differentials of lymphocyte and neutrophil, treatment modalities, and information regarding recurrence and death were collected from the medical records. Peripheral blood cell counts were used to calculate the absolute neutrophil count (ANC), absolute lymphocyte count (ALC), and NLR. The TNM stage was classified based on the American Joint Committee on Cancer, 7th edition, and the tumor grade followed the modified Scarf–Bloomer–Richardson grading system [39]. For the breast cancer subtyping, the following classifications were used: luminal/HER2 (−): ER positive and/or PR positive, and HER2 negative; HER2 (+): HER2 positive regardless of ER and PR status; and triple negative breast cancer (TNBC): ER negative, PR negative and HER2 negative.

Our study was approved by the Institutional Review Board (IRB) of the Gangnam Severance Hospital (Local IRB number: 3-2018-0341). The need for informed consent was waived by the approval of the IRB due to the retrospective design of the study. 

### 4.2. Treatment and Blood Cell Count Examination

Radiation was administered according to the following protocol: 50.4 Gy of radiation was administered in 28 fractions using X-ray linear accelerators (Elekta; Stockholm, Sweden) to the whole breast, followed by a boost dose of 9 Gy in 5 fractions delivered to the tumor bed. In cases with positive axillary lymph nodes or suspicious internal mammary lymph nodes, the total boost dose was increased to 10–16 Gy. All patients were treated with conventionally fractionated radiotherapy. Chemotherapy, anti-estrogen treatment, and anti-HER2 therapy were administered according to the standard protocols. 

Blood samples were collected at diagnosis, before the first (T1) and last (T2) RT administrations. T1 represents the baseline NLR, and the baseline blood work was done within two weeks before initiating RT. From then, routine blood work was performed every other week for monitoring purposes; T2 represents NLR results taken at the time point mostly before one week from the last session of RT. WBC counts were performed at the Department of Laboratory Medicine using an automated counting machine (Sysmex XN-series; Kobe, Japan). The Wright or Mary–Grinewald–Giemsa technique was used to determine the differential count; briefly, a drop of blood was thinly spread over a glass-slide, air dried, and stained with Romanowsky stain WBC differentiation was evaluated automatically using fractionation machines and manually in cases of morphological abnormalities. The absolute neutrophil and lymphocyte counts were calculated by multiplying each of their percentages with the total WBC count. The NLR was calculated by dividing the ANC by the ALC.

### 4.3. Statistical Analysis

In previous studies, the cut-off points for high NLR were not uniform. To determine the cut-off point for NLR in this study, all patients were randomly divided into training and validation sets in a ratio of 7:3. The reference sampling time was T2. The cut-off point for NLR in the training set was determined using the Youden index [40,41]. The discriminating ability was verified by the log-rank test in the validation set. The area under the curve (AUC) with ROC curves was used to determine the cut-off point. We assessed the optimal cut-off point for discriminating recurrence using the Youden index. The Youden index was calculated from the ROC curve as follows:

Youden index = max sensitivity − (1 − specificity).


Recurrence-free survival (RFS) was defined as the period from the date of surgery to the date of any loco-regional recurrence and/or distant metastasis of primary breast cancer. Overall survival (OS) was defined as the period from the date of primary surgery to the date of death due to any cause. Distant metastasis free survival (DMFS) was defined as the period from the date of primary surgery to the date of the distant metastasis. Data for patients who did not have an event of interest were censored at the date of the last follow-up.

Clinical characteristics were compared between groups with low and high NLR after RT. Patients who already had a high NLR before the administration of RT (based on the T1 counts) were excluded to solely evaluate the impact of RT and to preclude the effects of chemotherapy or surgery that were administered before RT. Continuous variables were compared between the two groups using the Student’s t-test or Mann-Whitney test. Categorical variables were compared using the Chi-square test or Fisher’s exact test. Survival curves were drawn using the Kaplan-Meier method and were compared between groups using the log-rank test. The uni- and multivariate Cox proportional hazard models were used to identify factors related to survival outcomes. The variables used in the multivariate analysis were those that showed statistical significance in the univariate analysis.

We used SPSS version 23 (SPSS; Chicago, IL, USA), R version 3.2.5 (www.R-project.org), and SAS version 9.4 (SAS Inc., Cary, NC, USA) software to perform these analyses. Statistical significance was defined as a *p*-value less than 0.05, and a 95% confidence interval (CI), not including 1.

## 5. Conclusions

Our results suggest that RT-induced high NLR is an adverse prognostic factor for breast cancer. It can be determined easily and reflects the patient’s immune status after RT, thus identifying patients at risk of tumor recurrence.

## Figures and Tables

**Figure 1 cancers-12-01896-f001:**
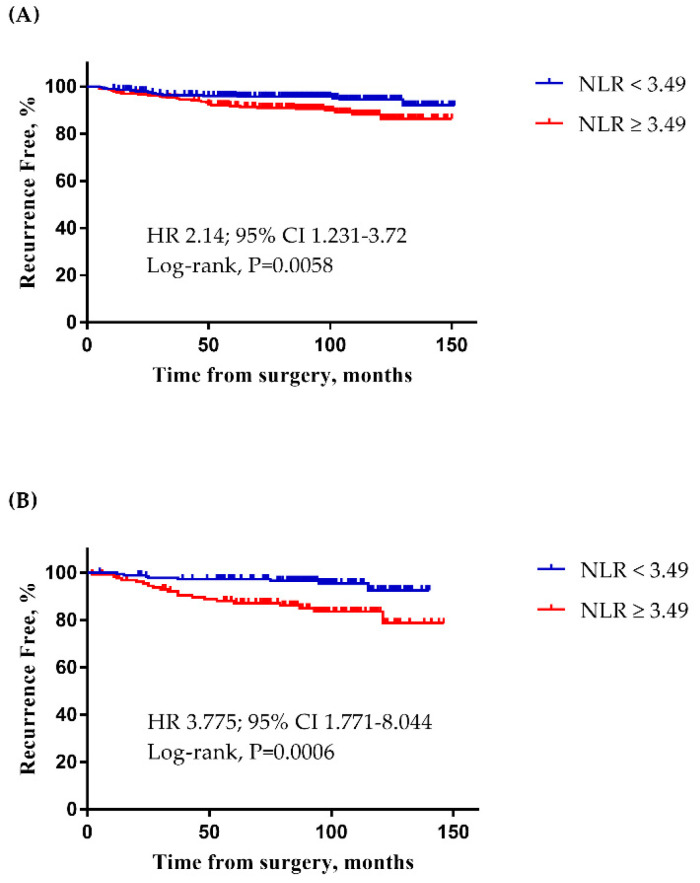
Kaplan-Meier survival curve of the neutrophil-to-lymphocyte ratio (NLR) cut-off point. (**A**) When the NLR cut-off point was 3.49 in the training set (*n* = 731), patients with high NLR showed lower recurrence-free survival (RFS; HR 2.14; 95% CI 1.231–3.72, *p* = 0.0058). (**B**) When the NLR cut-off point was 3.49 in the validation set (*n* = 313), patients with high NLR showed lower RFS (HR 3.775; 95% CI 1.771–8.044, *p* = 0.0006).

**Figure 2 cancers-12-01896-f002:**
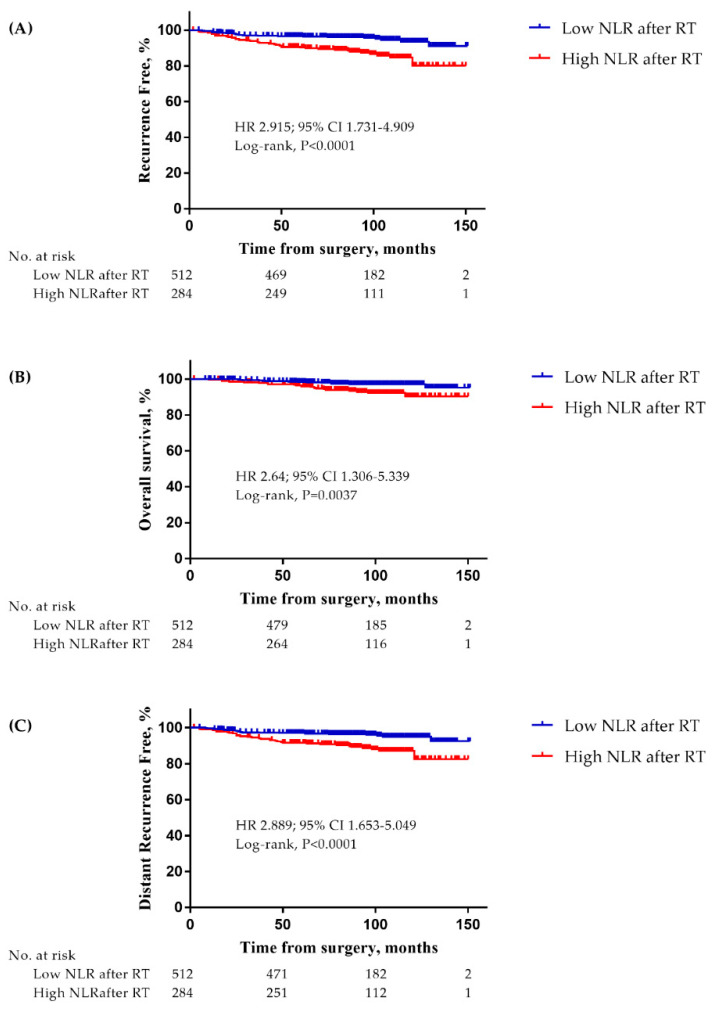
Kaplan-Meier survival curves of (**A**) RFS, (**B**) overall survival(OS), and (**C**) distant metastasis free survival (DMFS) based on high NLR after RT. (**A**) patients with high NLR after RT showed a lower RFS (HR 2.915; 95% CI 1.731–4.909, *p* < 0.0001), (**B**) patients with high NLR after RT showed a lower OS (HR 2.64; 95% CI 1.306–5.339, *p* = 0.0037), and (**C**) patients with high NLR after RT showed a lower DMFS (HR 2.889; 95% CI 1.653–5.049, *p* < 0.0001).

**Table 1 cancers-12-01896-t001:** Patients’ characteristics according to radiotherapy (RT) induced high and low neutrophil-to-lymphocyte ratio (NLR).

Variables	High NLR, *n* = 284 (%)	Low NLR, *n* = 512 (%)	*p* Value
Age, year			0.369
≥50	142 (50.0)	273 (53.3)	
<50	142 (50.0)	239 (46.7)	
ER			0.310
Positive	210 (73.9)	395 (77.1)	
Negative	74 (26.1)	117 (22.9)	
PR ^a^			0.463
Positive	169 (59.5)	317 (62.2)	
Negative	115 (40.5)	193 (37.8)	
HER2 ^a,b^			0.112
Positive	94 (33.3)	199 (39.0)	
Negative	188 (66.7)	311 (61.0)	
NG ^a^			0.176
I, II	194 (77.9)	329 (73.3)	
III	55 (22.1)	120 (26.7)	
HG ^a^			0.915
I, II	207 (76.1)	370 (76.4)	
III	65 (23.9)	114 (23.6)	
Tumor size ^a^, cm			0.968
≤2	239 (84.8)	430 (84.6)	
>2	43 (15.2)	78 (15.4)	
LN metastasis			0.001
Negative	225 (79.2)	449 (87.7)	
Positive	59 (20.8)	63 (12.3)	
Subtype ^a^			0.284
Luminal/HER2(−)	146 (51.8)	243 (47.8)	
HER2(+)	94 (33.3)	198 (39.0)	
TNBC	42 (14.9)	67 (13.2)	
Chemotherapy			0.003
Done	145 (51.1)	205 (40.0)	
Not done/unknown	139 (48.9)	307 (60.0)	
Regional nodal irradiation			<0.001
Done	104 (37.0)	75 (14.8)	
Not done	177 (63.0)	431 (85.2)	

Abbreviations: ER, estrogen receptor; PR, progesterone receptor; HER-2, human epidermal growth factor receptor-2; NG, nuclear grade; HG, histologic grade; LN, lymph node; TNBC, triple negative breast cancer AJCC stage was performed based on 7th edition; ^a^ Missing values are included, numbers may not sum to group totals; ^b^ HER-2 positive was defined by 3 positive on immunohistochemistry or amplification on fluorescence in situ hybridization.

**Table 2 cancers-12-01896-t002:** The hazard rations (HRs) and 95% confidence interval (CIs) for recurrence free interval (RFS) according to RT-induced high NLR.

Variables	Univariate Analysis		Multivariate Analysis	
	HRs (95% CIs)	*p* Value	HRs (95% CIs)	*p* Value
Age, year		0.946		
>50	1			
≤50	0.983 (0.597–1.618)			
NG		0.019		
I, II	1			
III	1.920 (1.114–3.308)			
HG		0.063		
I, II	1			
III	1.649 (0.974–2.792)			
ER		0.004		0.004
Negative	1		1	
Positive	0.469 (0.282–0.782)		0.451 (0.260–0.780)	
PR		0.019		
Negative	1			
Positive	0.551 (0.334–0.907)			
HER2		0.131		
Negative	1			
Positive	1.476 (0.890–2.447)			
Tumor size, cm		0.065		
≤2	1			
>2	1.754 (0.966–3.183)			
LN metastasis		<0.001		0.001
Negative	1		1	
Positive	4.439 (2.660–7.407)		2.774 (1.484–5.186)	
Subtype		0.066		
Luminal/HER2(−)	1			
HER2(+)	1.774 (1.014–3.104)			
TNBC	1.939 (0.965–3.896)			
RT-induced high NLR		<0.001		0.008
<3.49	1		1	
≥3.49	2.918 (1.750–4.866)		2.194 (1.230–3.912)	
Regional nodal irradiation		<0.001		0.052
Done	1		1	
Not done	3.869 (2.332–6.421)		1.879 (0.993–3.553)	
Chemotherapy		<0.001		
Done	1			
Not done/unknown	3.173 (1.853–5.435)			

Abbreviations: NG, nuclear grade; HG, histologic grade; ER, estrogen receptor; PR, progesterone receptor; HER-2, human epidermal growth factor receptor-2; LN, lymph node; TNBC, triple negative breast cancer; RT, radiotherapy; NLR, neutrophil to lymphocyte ratio.

**Table 3 cancers-12-01896-t003:** The HRs and 95% CIs for distant metastasis free survival (DMFS) according to RT-induced high NLR.

Variables	Univariate Analysis		Multivariate Analysis	
	HRs (95% CIs)	*p* Value	HRs (95% CIs)	*p* Value
Age, year		0.754		
>50	1			
≤50	0.918 (0.538–1.565)			
NG		0.076		
I, II	1			
III	1.681 (0.946–2.987)			
HG		0.108		
I, II	1			
III	1.593 (0.904–2.807)			
ER		0.082		
Negative	1			
Positive	0.605 (0.344–1.066)			
PR		0.102		
Negative	1			
Positive	0.640 (0.375–1.093)			
HER2		0.073		
Negative	1			
Positive	1.637 (0.955–2.805)			
Tumor size, cm		0.044		
≤2	1			
>2	1.898 (1.016–3.544)			
LN metastasis		<0.001		0.001
Negative	1		1	
Positive	4.452 (2.574–7.700)		2.860 (1.512–5.407)	
Subtype		0.191		
Luminal/HER2(−)	1			
HER2(+)	1.693 (0.951–3.014)			
TNBC	1.153 (0.492–2.698)			
RT-induced high NLR		<0.001		0.004
<3.49	1		1	
≥3.49	2.891 (1.673–4.998)		2.313 (1.299–4.120)	
Regional nodal irradiation		<0.001		0.078
Done	1		1	
Not done	3.452 (2.006–5.942)		1.777 (0.938–3.366)	
Chemotherapy		<0.001		
Done	1			
Not done/unknown	2.946 (1.665–5.213)			

Abbreviations: NG, nuclear grade; HG, histologic grade; ER, estrogen receptor; PR, progesterone receptor; HER-2, human epidermal growth factor receptor-2; LN, lymph node; TNBC, triple negative breast cancer; RT, radiotherapy; NLR, neutrophil to lymphocyte ratio.

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
