# Peer review of "Radiotherapy-Induced High Neutrophil-to-Lymphocyte Ratio is a Negative Prognostic Factor in Patients with Breast Cancer"

_cancers, 2020, doi:10.3390/cancers12071896_

Round 1
Reviewer 1 Report
The authors have adequately answered all queries arising from the previous draft of the manuscript and this is now acceptable for publication.
Reviewer 2 Report
The authors have adequately revised this manuscripts. Thank you!
This manuscript is a resubmission of an earlier submission. The following is a list of the peer review reports and author responses from that submission.
Round 1
Reviewer 1 Report
This study aimed to investigate the association between high neutrophil to lymphocyte ratio (NLR) after breast radiotherapy and survival outcomes. Overall, this is a well-written manusctipt with potential clinical relevance. My specfic comments are as follows:
1. Radiation has detrimental effects on bone marrow and there could contribute to myelosuppression. Breast radiotherapy after breast conservative surgery generally includes whole breast with or without regional nodal irradiation (supraclavicular or infraclavicular region, or internal mammary chain). Hence, small portions of ribs near the breast and clavicle could be irradiated and most of thoracic spines may not be irradiated during breast radiotherapy by current radiotherapy techniques. What's the number of patients were treated with whole breast radiotherapy only and whole breast radiotherapy with regional node irradiation, and the distribution of these patients between high-NLR and low-NLR groups? The authors should also analyse the association between thoracic bone marrow and the change in NLR. In an anlysis of distribution of whole body proliferating bone marrow, the ribs and clavicles only contained about 8.8% of proliferating bone marrow, while the thoracic spines contain about 20% and the lumbar spines and pelvic bones contain nearly 50%. I am not convinced enough about the radiation induced high neutrophil-lymphocyte ratio. Previous studies had revealed the association between hematologic toxicities and thoracic bone marrows in thoracic irradiation (Int J Radiat Oncol Biol Phys. 2016 Jan 1;94(1):147-54; Br J Radiol. 2016 Oct;89(1066):20160350). It would be better to evaluate the bone DVH data in your study for robustness.
2. Hypofractionated breast radiotherapy had become one of the standard of care in early stage breast cancer in current era. All of the patients receive 50.4 Gy at 1.8 Gy/fraction in your study. It would be nice to nice to explain or make some discussion about this issue.
3. A total of 248 patients with NLR ≥ 3.49 at T1 were excluded from the analysis. It is interesting what's the changes of NLR after radiotherapy and their outcomes. It would be nice to present their results in your study.
4. Chemotherapy is also myelosupressive in cancer patients. Breast cancer chemotherapy is associated with long-term changes in immune parameters that should be considered during clinical management. (Verma et al. Breast Cancer Res. 2016; 18: 10.) What's the types of chemotherapy the patients received, and the association between different types of chemotherapy and changes in NLR? What's the duration between the last chemotherapy and the blood T1 and T2, and their impact on NLR? The chemotherapy was also associated with high NLR in your Table 1. May chemotherapy also induce high NLR?
5. It is unknown why the authors used the reference saampling time at T2. The authors may need to describe more clear rationale in the methods. It is also unknown the cut-off of 3.49 would be easy to use in clinical practice. Previous studies had defined NLR cut-off at ≤3, 3-5, and >5 (Br J Cancer. 2018 Jul;119(1):40-51) based on patients with colon cancer though.
Reviewer 2 Report
The authors describe an interesting and novel study regarding the neutrophil to lymphocyte ration, and how this may represent a marker of systemic damage and an abnormal immune response following radiotherapy of breast cancer.
It would be useful to expand on the authors' thoughts on why the estrogen receptor status and lymph node metastases appears to correlate with NLR.
The abbreviation for ROC curve in the abstract should be explained.
The layout of the first paragraph of section 2.3 is not justified, in the same manner as other paragraphs.
The authors are aware of the limitations of the study and discuss them appropriately, however, the last sentence of Section 3 (Discussion) appears to be an unnecessary inclusion.
The acknowledgements section should be completed as appropriate.
The references in Materials and Methods section are not presented as references in the rest of the manuscript.